# TMT-Based Quantitative Proteomics Reveals Cochlear Protein Profile Alterations in Mice with Noise-Induced Hearing Loss

**DOI:** 10.3390/ijerph19010382

**Published:** 2021-12-30

**Authors:** Long Miao, Juan Zhang, Lihong Yin, Yuepu Pu

**Affiliations:** Key Laboratory of Environmental Medicine Engineering, Ministry of Education, School of Public Health, Southeast University, Nanjing 210009, China; miaolong0308@163.com (L.M.); 101011288@seu.edu.cn (J.Z.); lhyin@seu.edu.cn (L.Y.)

**Keywords:** noise-induced hearing loss, proteomics, cochlea, autophagy, inflammation

## Abstract

Noise-induced hearing loss (NIHL) is a global occupational disease affecting health. To date, genetic polymorphism studies on NIHL have been performed extensively. However, the proteomic profiles in the cochleae of mice suffering noise damage remain unclear. The goal of this current study was to perform a comprehensive investigation on characterizing protein expression changes in the cochlea based on a mouse model of NIHL using tandem mass tag (TMT)-labeling quantitative proteomics, and to reveal the potential biomarkers and pathogenesis of NIHL. Male C57BL/6J mice were exposed to noise at 120 dB SPL for 4 h to construct the NIHL mouse model. The levels of MDA and SOD, and the production of proinflammatory cytokines including TNF-α and IL-6 in the mice cochleae, were determined using chemical colorimetrical and ELISA kits. Moreover, differentially expressed proteins (DEPs) were validated using Western blotting. The mouse model showed that the ABR thresholds at frequencies of 4, 8, 12, 16, 24 and 32 kHz were significantly increased, and outer hair cells (HCs) showed a distinct loss in the noise-exposed mice. Proteomics analysis revealed that 221 DEPs were associated with NIHL. Bioinformatics analysis showed that a set of key inflammation and autophagy-related DEPs (ITGA1, KNG1, CFI, FGF1, AKT2 and ATG5) were enriched in PI3K/AKT, ECM-receptor interaction, and focal adhesion pathways. The results revealed that the MDA level was significantly increased, but the activity of SOD decreased in noise-exposed mice compared to the control mice. Moreover, TNF-α and IL-6 were significantly increased in the noise-exposed mice. Western blotting revealed that the expression levels of ITGA1, KNG1, and CFI were upregulated, but FGF1, AKT2, and ATG5 were significantly downregulated in noise-exposed mice. This study provides new scientific clues about the future biomarkers and pathogenesis studies underlying NIHL. Furthermore, the findings suggest that the validated DEPs may be valuable biomarkers of NIHL, and inflammation and autophagy may be pivotal mechanisms that underlie NIHL.

## 1. Introduction

Noise is a widespread occupational hazard affecting the health of workers. Noise-induced hearing loss (NIHL) is a progressive sensorineural hearing loss caused by prolonged noise exposure. NIHL is a major occupational-related disorder and is a hard occupational health issue to solve in workplace safety [1]. It is worth noting that approximately 10% of the world’s population is permanently exposed to high-intensity noise and are at risk of suffering NIHL [2]. NIHL has been the third largest occupational health problem in China, the incidence of which has been increasing [3].

NIHL is an extremely complex neurological hearing impairment caused by a variety of environmental and genetic factors. Oxidative stress is thought to be involved in the occurrence and development of NIHL by inducing hair cells (HCs) loss in the inner ear [4]. Oxidative damage is mainly caused by the production of increased reactive oxygen species (ROS) or decreased antioxidant activity. Available evidence showed that overproduction of ROS has become a frequent pathogenesis of a variety of functional injuries in the inner ear caused by ototoxic drug therapy and noise exposure [5]. In addition, our recent findings found that inflammation-related genes polymorphisms are significantly associated with NIHL, suggesting that inflammation may be an important pathogenesis contributing to cochlear impairment induced by noise exposure [6]. In an increasing number of studies were conducted, however, the concrete mechanisms of NIHL were equivocal. Thus, the identification of novel physiopathological candidate proteins is significant to reveal the pathogenesis, early diagnosis, and therapeutic targets of NIHL.

Proteomics has powerful capacity to identify the potential biomarkers and therapeutic targets and it has been increasingly used to screen biomarkers for various diseases. In the last few years, proteomics has been performed to explore differentially expressed proteins (DEPs) in dengue fever [7], diabetes [8], Alzheimer’s disease [9], and pulmonary fibrosis induced by PM_2.5_ [10]. There are several proteomics methods used for tagged or isotope labeled quantification including tandem mass tag (TMT) and isobaric tags for relative and absolute quantitation (iTRAQ) with the advantages of high sensitivity and reproducibility [11,12]. However, as far as we know, little is known about the proteomics studies of NIHL.

To explore and identify potential protein biomarkers for prediction, diagnosis, and treatment of NIHL, TMT-labeled coupled with ultra-high performance liquid chromatography tandem mass spectrometry (UPLC-MS/MS) quantitative proteomics was employed to characterize DEPs of cochlear tissue based on an animal model of NIHL using C57BL/6J strain mice. In addition, this study will provide comprehensive insights into physiopathological alterations of NIHL and enhance understanding of potential molecular mechanisms underlying the development of NIHL.

## 2. Materials and Methods

### 2.1. Animals and Noise-Induced Hearing Loss (NIHL) Model Construction

The flow chart of this study design is shown in Figure 1. Male C57BL/6J mice aged 6–8 weeks old and weighing 18–24 g with normal hearing function were obtained from Jackson Laboratories and housed with free access to regular rodent chow and distilled water under pathogen-free standard conditions (23 °C, 50% relative humidity, 12 h light/dark cycle). Only male mice were used in this present study because the estrous cycle of female mice may affect the acoustic startle reflex (ASR) amplitude. Previously, it has been reported that mice exposed to white noise at 120 dB sound pressure level (SPL) for 2 h experienced a significant NIHL and sensory HCs loss in the cochlea [13]. In this study, we constructed an NIHL model by exposing C57BL/6J strain mice to broadband white noise under 120 dB SPL for 4 h condition to induce severe NIHL. A total of 36 mice were used and randomly divided into the noise group (*n* = 18) and control group (*n* = 18).

Noise exposure was generated through white noise synthesized using a digital signal processor (Intelligent Hearing Systems, Miami, FL, USA) joined to a personal computer and custom designed noise filter (Intelligent Hearing Systems, Miami, FL, USA). Then, the noise signal was amplified and transmitted by speakers (RadioShack Corp, Fort Worth, TX, USA) mounted on the walls of the acoustic chamber. During noise exposure, each mouse was placed into a custom-designed metal mesh cage and placed in the center of the acoustic chamber. Noise exposure intensity was measured byTES-1350A sound level meter (TES, Taiwan). However, the gender and age-matched control mice were arranged in a silent room without noise exposure. All experiments were performed according to the protocols approved by the Animal Care and Use Committee of Zhongda Hospital, Affiliated to Southeast University (approval No: 2020ZDSYLL150-P01). In addition, all animal experiments in this study were in accordance with the National Institutes of Health guide for the care and use of laboratory animals. Furthermore, all efforts were made to reduce the number of animals used and to protect them from suffering.

### 2.2. Auditory Brainstem Response (ABR) Measurement

After noise exposure, all mice in noise-exposed group and control group were examined for auditory brainstem response (ABR) measurements in a sound-isolated room by TDT system III equipment (Tucker-Davies Technologies, Gainesville, FL, USA). Mice were deeply anesthetized by intraperitoneal injection using pentobarbital. Stainless-steel subdermal needle electrodes were separately fixed at the vertex of the head (active electrode), right mastoid (reference electrode) and base of tail (ground electrode). Tone burst at the frequencies of 4, 8, 12, 16, 24, and 32 kHz was transmitted to test the external auditory canal by means of a miniature earphone. To obtain auditory thresholds, the sound intensity of the tone burst started at 90 dB SPL and decreased by 10 dB intervals to determine the hearing threshold. The elicited response was amplified and scanned 1024 times on average in real time. Auditory threshold was defined as the lowest sound intensity at which an obvious ABR wave could be observed [14]. 

### 2.3. Cochlea Surface Preparations and Hair Cells (HCs) Counting

Briefly, three days after ABR measurement, three mice for each group were anesthetized and the cochleae were extracted and then put into 0.1 M phosphate-buffered saline (PBS) (catalog no. P5244, Sigma-Aldrich, Saint Louis, MO, USA), and then 4% paraformaldehyde (catalog no. 158127, Sigma-Aldrich, Saint Louis, MO, USA) was perfused and incubated at 4 °C overnight. The cochleae were washed in PBS and decalcified by 10% ethylenediamine tetraacetic acid (EDTA) for 3 days at 4 °C. After incubation in a 10% goat serum for blocking non-specific antibody binding overnight at 4 °C, cochleae were incubated at 4 °C overnight in the darkness using an anti-Myosin7a antibody (catalog no. ab3481, Abcam, MA, USA) at 1:100 dilution. After washing in PBS, cochleae were incubated with phalloidin conjugated with fluorescent dyes (Alexa Fluor™ 488, catalog no. A12379, Life technology, CA, USA) for 2 h at room temperature in darkness. The cochleae were dissected into equally sized apex, middle, and base and fixed on slides using Fluoromount-G mounting medium (catalog no. 0100-01, Southern Biotech, Birmingham, AL, USA). Images were processed and obtained by a Zeiss microscope at 40× magnification from the apex through the base of cochlear surface preparations. The numbers of immune-stained outer hair cells (OHCs) and inner hair cells (IHCs) were quantified per 100 μm length of cochlea in all three segments using ImageJ (National Institutes of Health, Bethesda, MD, USA). 

### 2.4. Protein Extraction and Digestion 

Following ABR measurement, six mice each of the control group and noise-exposed group were immediately decapitated for further proteomics analysis, both temporal bones were separated and positioned in ice-cold Hanks’ balanced salt solution (HBSS) (catalog no. H4641, Sigma-Aldrich, Saint Louis, MO, USA). The cochlear bone capsule was excised with a 28 G needle tip under an anatomical microscope (Stemi DV4, Carl Zeiss, Germany) in accordance with the standard anatomical method. Basilar membrane, spiral ligament, along with stria vascularis were separated. In total, four cochleae from two mice were mixed as a sample and then homogenized in ice-cold RIPA lysis buffer containing phosphatase and protease inhibitors (catalog no. 89900, Thermo Fisher Scientific, Waltham, MA, USA) using a glass pellet pestle. After homogenization on ice for 1 min, the samples were processed by ultrasound with a Scientz-IID ultrasonic processor (Scientz, Ningbo, China) for 10 min on ice, and then were centrifuged at 12,000× *g* for 10 min at 4 °C. The supernatant was reserved, and concentration was determined by BCA assay kit (catalog no. P0010, Beyotime Biotechnology, Shanghai, China). The protein solution was reduced using 5 mM DL-dithiothreitol (DTT) (catalog no. A100281, Sangon Biotech, Shanghai, China) at 55 °C for 30 min, then alkylated by 10 mM iodoacetamide at room temperature for 15 min in darkness. The protein sample was redissolved in 200 mM triethylammonium bicarbonate (TEAB) (catalog no. 90114, Thermo Fisher Scientific, Waltham, MA, USA). Eventually, protein samples were digested using trypsin at a mass ratio of 1:50 (trypsin to protein) at 37 °C overnight. 

### 2.5. Tandem Mass Tag (TMT) Labeling and HPLC Fractionation

After digestion, the samples were redissolved in 50 μL TEAB (100 mM). TMT labeling 6-plex kits (catalog no. 90064CH, Thermo Fisher Scientific, Waltham, MA, USA) were separately dissolved in 88 μL HPLC-grade acetonitrile (Thermo Fisher Scientific, Waltham, MA, USA). Each sample with 50 μL TEAB buffer was mixed with 41 μL of prepared TMT reagent at room temperature for 1 h. The reaction was then stopped by 8 μL 5% hydroxylamine (Sigma-Aldrich, Saint Louis, MO, USA). Finally, TMT-labeled peptides were dried by vacuum centrifugation. 

TMA-labeled peptide mixtures were fractionated according to high pH reverse-phase HPLC method with an Agilent Zorbax Extend C18 column (5 μm, 150 mm × 2.1 mm). Mobile phase was made up of water with 2% acetonitrile (A) and 90% acetonitrile in water (B). Procedures of solvent gradient elution were as follows: 98% A for 0–8 min, 98–95% A for 8–8.01 min, 95–75% A for 8.01–48 min, 75–60% A for 48–60 min, 60–10% A for 60–60.01 min, 10% A for 60.01–70 min, 10–98% A for 70–70.01 min, and 98% A for 70.01–75 min. The peptides were separated at a delivery flow rate of 0.3 mL/min and detection wavelength was set, respectively, at 210 nm and 280 nm. Samples were collected from 8 to 60 min and the elution buffers were harvested into the centrifuge tube numbered from 1 to 15 at interval of 1 min. After collection, the separated peptides were freeze-dried in vacuum for further analysis.

### 2.6. Liquid Chromatography-Tandem Mass Spectrometry (LC-MS/MS) Analysis

LC-MS/MS analysis was conducted using Triple TOF 5600 System (AB SCIEX, Framingham, MA, USA). The mobile phases were water with 0.1% formic acid (A) and 99.9% acetonitrile with 0.1% formic acid (B). Chromatographic separation was carried out on a reverse-phase C18 column (3 μm, 15 cm × 75 μm). The delivery flow rate was set at 1 μL/min. 

The parameters were as follows: ion spray voltage, 1.5 kV; curtain gas, 35 psi; nebulizer gas, 5 psi; interface heater temperature, 180 °C; collision energy, 20–59 eV. MS scanning range was 100–1700 *m*/*z* for data acquisition. 

### 2.7. Database Search and Bioinformatics Analysis

The original data were processed using ProteinPilot software package. Detailed parameters were set as follows: fixed modification of carbamidomethyl on cysteines, variable modification of oxidation on methionine, instrument type Bruker TIMS, trypsin digestion, first search peptide tolerance of 20 ppm, main search peptide tolerance of 10 ppm, max 2 missing cleavages, and false discovery rate (FDR) less than 1%. In addition, a *p* value of <0.05 was considered as statistically significant. Proteins with *p* value < 0.05 and fold change of >1.20 or <0.83 were considered as significantly up-regulated or down-regulated proteins.

Gene ontology (GO) annotation including biological process (BP), cellular compartment (CC) and molecular function (MF) were analyzed using GO database (http://geneontology.org, accessed on 1 February 2021). The significance of GO enrichment was defined as *p* < 0.05. Pathway enrichment analysis of DEPs was carried out using Kyoto Encyclopedia of Genes and Genomes (KEGG) online database (https://www.genome.jp/kegg/mapper/, accessed on 1 February 2021) and *p* value < 0.05 was considered as of statistical significance. Protein–protein interaction (PPI) analysis was analyzed by the STRING database (https://string-db.org, accessed on 30 July 2021).

### 2.8. Oxidative Stress Markers Levels Assay, and Proinflammatory Cytokines Measurement

After noise exposure, mice cochleae were collected and processed by ultrasound with a Scientz-IID ultrasonic processor (Scientz, Ningbo, China) for 10 min on ice and centrifuged at 12,000× *g* at 4 °C for 10 min. The supernatant was reserved at −80 °C until analysis. Superoxide dismutase (SOD) activity (catalog no. A001-3, Jiancheng Bioengineering, Nanjing, China) and level of malondialdehyde (MDA) (catalog no. A003-1, Jiancheng Bioengineering, Nanjing, China) were determined using commercial chemical colorimetrical kits according to the manufacturer’s instructions. Moreover, proinflammatory cytokines including TNF-α (catalog no. AF-02415M2, AiFang biological, Changsha, China) and IL-6 (catalog no. AF-02446M2, AiFang biological, Changsha, China) were determined by ELISA kits. A sample size of three mice was used for each group.

### 2.9. Western Blot Analysis

Equal amounts of protein samples were separated using 12.5% sodium dodecyl sulfonate-polyacrylamide gel electrophoresis (SDS-PAGE) and transferred to polyvinylidene difluoride membranes. Then, the membranes were blocked at room temperature for 2 h with 5% non-fat milk in a TBST buffer and incubated with primary antibodies against ITGA1 (1:1000, DF2538, Affinity Biosciences, Cincinnati, OH, USA), KNG1 (1:1000, A1670, ABclonal Technology, Wuhan, China), CFI (1:1000, A5623, ABclonal Technology, Wuhan, China), FGF1 (1:1000, DF6124, Affinity), AKT2 (1:1000, AF6264, Affinity), ATG5 (1:1000, DF6010, Affinity Biosciences, Cincinnati, OH, USA) and GAPDH (1:1000, ab8245, Abcam Cambridge, MA, USA) at 4 °C overnight, then incubated for 2 h at room temperature by HRP goat anti-rabbit IgG antibody (1:5000, AS014, ABclonal Technology, Wuhan, China). The protein was visualized by Tanon-5200 Chemiluminescent Imaging System (Tanon Science & Technology, Shanghai, China). Three independent assays were implemented.

### 2.10. Statistical Analysis

Continuous variables with normal distribution were expressed as mean ± standard deviation (SD). Differences in the expression level of proteins, SOD, MDA, TNF-α, and IL-6 between noise-exposed and control mice were analyzed. Independent-samples student’s *t*-test was used for data analysis. Statistical analysis was conducted by SPSS 23.0 software (SPSS, Chicago, IL, USA). *p* < 0.05 was considered significance. *p* value was indicated by asterisks (* *p* < 0.05, ** *p* < 0.01, *** *p* < 0.001).

## 3. Results 

### 3.1. ABR Thresholds and HCs Immunofluorescence Staining and Counting

C57BL/6J mice were used to construct a mouse model of NIHL under 120 dB 4 h noise exposure condition. The effects of noise on ABR thresholds are shown in Figure 2a. The results showed that the ABR mean thresholds of the noise-exposed mice group (87.92 ± 2.57 at 4 kHz, 83.75 ± 3.11 at 8 kHz, 80.42 ± 3.34 at 12 kHz, 81.67 ± 3.26 at 16 kHz, 87.50 ± 4.52 at 24 kHz, 88.75 ± 3.11 at 32 kHz) were significantly higher than those of the control mice (44.17 ± 4.69 at 4 kHz, 27.92 ± 4.98 at 8 kHz, 22.08 ± 2.57 at 12 kHz, 30.42 ± 3.34 at 16 kHz, 38.33 ± 4.44 at 24 kHz, 51.25 ± 3.11 at 32 kHz) at all tested frequencies, showing that noise-exposed mice developed to a severe NIHL (*p* < 0.001).

To determine whether the noise-exposed mice having a typical NIHL with the HCs loss, the pattern of HCs was further evaluated. Myosin7a and phalloidin immunofluorescence staining showed that there was no IHCs deficiency, however, OHCs showed scattered loss and exhibited degeneration and were disorganized in noise-exposed mice, especially in the middle and basal segments of the cochlea. Conversely, both OHCs and IHCs of the control mice appeared to have a normal pattern with a distinct structure and no degeneration (Figure 2b). Besides, OHCs loss was significantly increased in noise-exposed mice compared to control mice (Figure 2c, *p* < 0.01). The results revealed that 120 dB SPL 4 h noise exposure condition could cause severe NIHL and impair the OHCs.

### 3.2. Identification of Differentially Expressed Proteins (DEPs)

To explore the protein expression changes, TMT-based quantitative proteomics analysis was applied to characterize DEPs in the cochleae from noise-exposed and control mice. A total of 4435 proteins were quantified with a false discovery rate (FDR) <1%, of which 4431 proteins having at least 1 unique peptide. According to the following screening criteria of *p* value < 0.05 and 1.20-fold change, 221 of the 4431 proteins were regarded as the significant DEPs. Of these DEPs, 110 were up-regulated proteins, and 111 were down-regulated proteins. A clustering map was further construct to clearly show the differences in protein abundance between the control group and noise group (Figure 3). The results showed that the trend of DEPs exhibited a good consistency between the three samples in each group.

### 3.3. Functional Enrichment Analysis of DEPs

To understand the effects of noise on the cochlea and explore the functional characteristics of DEPs, GO analysis was performed on the overall functional properties based on BP, CC, and MF. As shown in Figure 4a, the DEPs were mainly involved in processes of the innate immune response, extracellular matrix organization, and defense response to virus. The DEPs were found to be located in the extracellular region, collagen-containing extracellular matrix, and extracellular matrix. Moreover, we found that the major MFs of DEPs were related to the identical protein binding, protein homodimerization activity, protease binding, and extracellular matrix structural constituent.

### 3.4. Pathway Enrichment and Protein–Protein Interaction (PPI) Network Analysis

To investigate the signal transduction pathways that DEPs may participate in, KEGG pathway analysis was conducted. DEPs were significantly involved in PI3K/AKT signaling pathway, ECM-receptor interaction, focal adhesion, protein digestion and absorption, complement and coagulation cascades, platelet activation, neutrophil extracellular trap formation, nitrogen metabolism, longevity-regulating pathway, selenocompound metabolism, and glycolysis/gluconeogenesis (Table 1). 

PPI network analysis of DEPs participated in the significant signaling pathways was performed. As displayed in Figure 4b, one group of DEPs (COL1A1-COL1A2-COL2A1-COL9A2-COL11A1-COL5A2-COL11A2-COL28A1-ITGA1-COL6A5) interacted strongly and was involved in the PI3K/AKT signaling pathway, ECM-receptor interaction, as well as focal adhesion. The DEPs, such as VWF, A2M, SERPINA1B, SERPINA1E, SERPINA1A, KNG1, FGB, PLG, FGA and FGG were found to have strong interaction with each other and to participate in complement and coagulation cascades process. There were possible direct and indirect interactions among the rest of the DEPs. 

### 3.5. Validation of DEPs by Western Blot Analysis 

Further analysis revealed that DEPs involving in the significant signaling pathways consisted of some proteins (ITGA1, KNG1, CFI, FGF1, AKT2, and ATG5) closely related to autophagy and inflammatory response. Our recent studies indicated that autophagy and inflammatory response may be a key pathological mechanism involving in NIHL [15]. Thus, these six DEPs were selected for validation (Table 2). The results showed that the expression levels of FGF1 (0.74-fold), AKT2 (0.83-fold) and ATG5 (0.64-fold) were significantly decreased in noise-exposed mice. In contrast, compared to the control mice, the expression of ITGA1 (1.63-fold), KNG1 (1.24-fold), and CFI (1.16-fold) were upregulated in noise-exposed mice (Figure 5a). The results were in accordance with the findings obtained from the TMT-labeled quantitative proteomics analysis. Taken together, the findings suggested that these DEPs may be important molecules for NIHL, and autophagy and inflammation probably play a noticeable role in the process of NIHL.

### 3.6. Measurement of Superoxide Dismutase (SOD) Activity, Malondialdehyde (MDA) Level and Proinflammatory Cytokines Production

Oxidative stress and inflammation response play an important role in the NIHL. SOD and MDA, two important oxidative stress markers were selected to measure the oxidative damage in the mice cochleae of the two groups. The results indicated that the SOD activity was significantly decreased in noise-exposed mice relative to the control mice. However, the MDA level in noise-exposed mice was significantly increased compared to the controls (Figure 5b). Therefore, the findings suggested that noise exposure may cause the increase of MDA level and decrease of SOD activity in the cochlea. 

In addition, proinflammatory cytokines production of TNF-α and IL-6 were further determined in the cochleae of noise-exposed and control mice. Compared with the control mice, TNF-α and IL-6 showed a significant increase in the noise-exposed mice, revealing that noise might trigger inflammation involved in cochlear pathogenesis (Figure 5c).

## 4. Discussion

NIHL is a severe occupational-related disease and occupational health issue worldwide [1]. In this study, cochlear proteomic characteristics of the noise-exposed mice and control mice were analyzed and compared using the TMT-based quantitative proteomics method based on the NIHL mouse model. After exposure to 120 dB SPL noise for 4 h, the noise-exposed mice developed a typical consequence of NIHL with increased of ABR thresholds at 4, 8, 12, 16, 24, 32 kHz and loss of OHCs in the cochlea, compared to the control mice. Proteomics analysis revealed that a set of key inflammation and autophagy-related DEPs (ITGA1, KNG1, CFI, FGF1, AKT2, and ATG5) were enriched in the signaling pathways such as PI3K/AKT pathway, ECM-receptor interaction focal adhesion pathway. Western blotting analysis suggested that the expression of ITGA1, KNG1 and CFI were significantly upregulated but FGF1, AKT2, and ATG5 expression were decreased after noise exposure, showing that autophagy and inflammation may play an important role in NIHL. This study demonstrates the proteomics response to noise exposure in mice cochleae and identifies protein biomarkers correlated with NIHL. Furthermore, this study provides a theoretical basis for further studies on the pathogenesis of NIHL. 

Previous studies have revealed that ROS was a non-negligible pathological molecular mechanism involved in inner ear injury induced by noise exposure [5,16]. Excessive accumulation of ROS could cause antioxidant dysfunction, membranes and protein impairment, and triggered vasoconstriction and ischemia/reperfusion injury, thereby leading to oxidative damage in the cochlea during noise exposure [13,17]. Moreover, the appearance of ROS was an early phenomenon in the HCs damage process arise from high-intensity noise exposure, indicating that ROS may be a key factor involved in the initiation of cochlear injury [18,19]. It has been suggested that several enzymes including SOD and GSH could protect cells from the damage of oxygen free radicals [20]. MDA is the product of oxygen free radicals, which could effectively reflect the damage of ROS [21]. We found that the MDA level significantly increased, and SOD activity obviously decreased after noise exposure, indicating that NIHL was associated with oxidative damage and ROS-mediated inflammatory response. In addition, similar results were also presented in previous studies [22,23]. In addition to oxidative damage, available evidence has indicated that proinflammatory cytokines and chemokines increased in cochleae with damage induced by noise exposure [24,25]. Both TNF-α and IL-6, are important inflammatory molecules and considered to be involved in hearing damage [26,27]. Our results showed that the levels of TNF-α and IL-6 were significantly upregulated in noise-exposed mice compared to the control mice. Previously, one animal study found that noise exposure could significantly increase the production of pro-inflammatory cytokines such as TNF-α, IL-6, and IL-1β in the cochleae [28]. Arslan et al. performed animal experiment and observed that the level of TNF-α cytokine was significantly higher in noise-exposed rats cochleae compared to the untreated control rats [29]. Moreover, another animal experiment found that mice with permanent hearing threshold shift (PTS) showed significantly higher levels of IL-1β, IL-6, and TNF-α in the cochlea [30]. Beyond mouse experiments, TNF-α and IL-6 levels were significantly higher in the perilymph fluid of humans with deafness [31]. Our current findings were consistent with the above findings, revealing that inflammation may be an essential factor contributing to the pathogenesis of NIHL.

In this current study, we found that a set of DEPs related to autophagy and inflammation including three upregulated DEPs (ITGA1, KNG1 and CFI) and three downregulated DEPs (FGF1, AKT2, and ATG5) were involved in the PI3K/AKT pathway, ECM-receptor interaction, and focal adhesion pathway signaling pathways and may be important biomarkers of NIHL. Our previous studies reported that autophagy and inflammation may be noticeable pathological mechanisms involved in the risk of NIHL [15,32]. Western blotting suggested that the expression levels of FGF1, AKT2, and ATG5 were decreased, but the ITGA1, KNG1 and CFI expression levels were upregulated in noise-exposed mice, compared to control mice. ITGA1 encodes the α1 subunit of integrin receptors and plays a vital role in inflammation and fibrosis [33]. ITGA1 is an important signaling molecule involved in the progress of regulating apoptosis, gene expression, cell proliferation, invasion, metastasis, and angiogenesis [34]. Yim et al. revealed that ITGA1 was involved in the adhesion of gastric cancer cells to peritoneum and may have an important role in gastric cancer [35]. ITGA1 was also reported to be closely associated with diabetes and diabetic retinopathy [34,36]. A previous study has revealed that abnormal expression of ITGA1, which was an inflammatory-associated gene, may be associated with the development of autoimmune inner ear disease [37]. In this study, ITGA1 was found to be significantly upregulated in the noise-stimulated mice cochlea. ITGA1 is a critical protein involved in inflammation response, and it has been shown that ITGA1 deletion increases the exit efficiency of macrophages, which are key modulators during inflammation [38]. The increased expression level of ITGA1 during cochlear inflammation may be an adaptive response to prevent sensory HC injury induced by inflammatory mediators. MacDonald et al. has reported that loss of epithelial integrity caused by a decreased function of integrin may lead to more severe injury of the epithelium [39]. The specific functional role of ITGA1 in NIHL needs further validation. 

Langhauser et al. found that the absence of KNG1 could significantly decrease thrombosis and inflammation in ischemic mice [40]. Moreover, as a pro-inflammatory cytokine, KNG1 was found to aggravate the progress of inflammation, and inhibition of KNG1 could mitigate inflammation in the cells [41,42]. The results of this current study showed that KNG1 expression was significantly upregulated in cochlear tissues following noise exposure, suggesting that there was an underlying regulatory relationship between KNG1 and NIHL. The CFI gene encodes a serine proteinase that is essential for regulating the complement cascade, and is widely expressed in macrophages [43]. CFI-mediated complement regulation was reported to be involved in the progression of Alzheimer’s disease [44]. Moreover, it was reported that CFI deficiency was related to cerebral inflammation [45]. In this study, the expression level of CFI protein in the cochlea was upregulated. The results may suggest that CFI might involve mediating the immune response to noise stress. Moreover, FGF1 involving in PI3K/AKT pathway is a key modulator for endothelial cell migration, proliferation, and an angiogenic factor [10]. FGF1 was also associated with the development and progression of arthritis [46]. The expression of FGF1 in noise-exposed mice was significantly decreased. A vital function of FGF1 is the promotion of endothelial cell proliferation thus promoting angiogenesis [47]. Taken together, we could speculate that the abnormal FGF1 expression impairs the endothelial cell proliferation in the cochlea, thus inhibiting angiogenesis and promoting NIHL. Nonetheless, the specific mechanisms of CFI and FGF1 proteins in the development of NIHL still need to be further studied. We also found that the expression of AKT2 and ATG5 decreased in NIHL mice compared to control mice. Our recent study observed that the mRNA expression levels of AKT and ATG5 were significantly decreased in NIHL workers compared with the normal hearing workers, suggesting that autophagy and inflammation probably were noticeable in the pathological processes of NIHL. PI3K/AKT was considered a key regulator for cell growth and survival and has a critical role in mediating oxidative stress [48]. Available evidence showed that PI3K/AKT was correlated with inner ear sensory HC death caused by multifarious external injuries and stimuli [49]. Moreover, recent research generated mice deficient in ATG5 and found that deletion of ATG5 could lead to HCs degeneration and severe congenital hearing loss. Taken together, the current findings showed that autophagy and inflammation may play critical roles in the etiology of NIHL. 

There were some limitations that deserve discussion in this present study. For proteomics analysis, three mixed samples per group were analyzed, and the small size of samples may result in incomplete or inaccurate results of proteomics. Further larger sample studies are required to verify whether the identified DEPs correlate with NIHL. Moreover, due to the low number of samples used for the measurement of oxidative stress and pro-inflammatory cytokine, and Western blotting analysis, the study may be underpowered to show the real levels of oxidative stress, cytokine, and DEPs in the damaged cochlea caused by noise. Therefore, this exploratory approach needs to be validated on large independent cohorts. In addition, functional studies are necessary to confirm and elucidate the potential mechanisms of these DEPs involved in NIHL. Lastly, due to the large number of mice used, the mice were separated for multiple noise exposure, which may lead to experiment errors. Future studies should focus on subjecting mice to a concentrated noise exposure and then explore the protein expression changes in the cochlea.

## 5. Conclusions

The NIHL mouse model suggested that a noise exposure condition of 120 dB SPL for 4 h could cause significantly increased ABR thresholds and OHC loss in the cochlea. The MDA level was significantly increased, but SOD activity was decreased. Moreover, pro-inflammatory cytokines of TNF-α and IL-6 were significantly increased in the mice cochleae after noise exposure, indicating that oxidative stress and inflammatory response occurred and these may be essential underlying mechanisms for the development of NIHL. Moreover, several important autophagy and inflammation-related DEPs (ITGA1, KNG1, CFI, FGF1, AKT2 and ATG5) were confirmed to be associated with NIHL. Following noise exposure, aberrant expression of these proteins may reveal the underlying mechanism of NIHL and show autophagy and inflammation may be critical roles in the etiology of NIHL. Furthermore, this present study provides a new perspective for the progress of NIHL and offers new clues about the mechanisms underlying NIHL. Further studies are needed to confirm the findings and to explore the mechanism of these DEPs contributing to NIHL.

## Figures and Tables

**Figure 1 ijerph-19-00382-f001:**
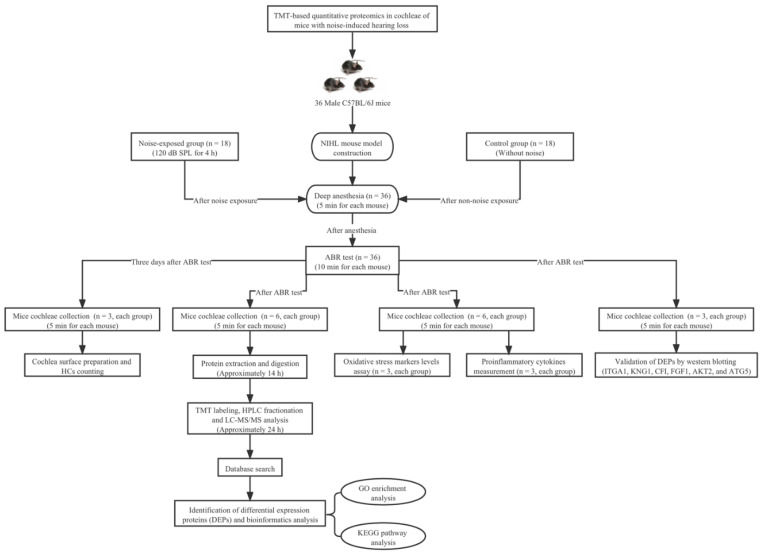
The flow chart of this study design.

**Figure 2 ijerph-19-00382-f002:**
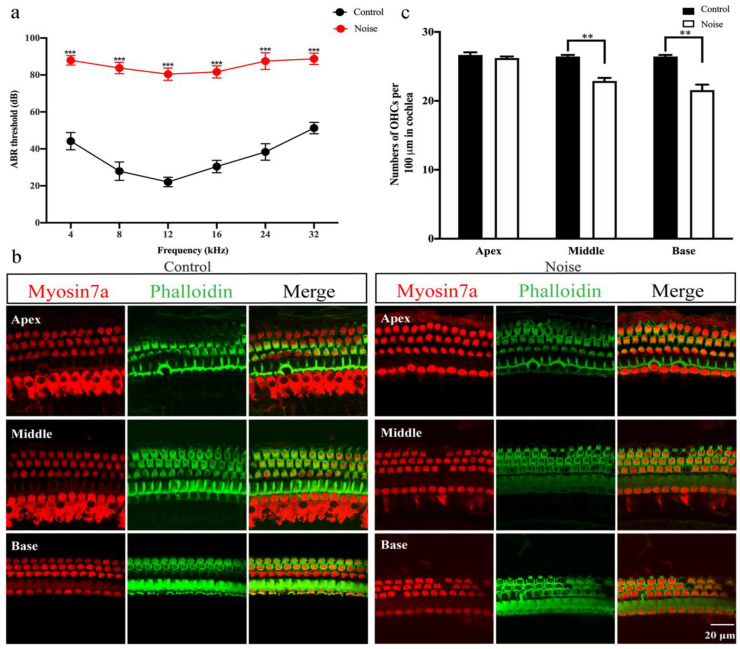
Noise-induced hearing loss (NIHL) mouse model was constructed under 120 dB SPL for 4 h condition. (**a**) Comparison of mean auditory brainstem response (ABR) thresholds at frequencies of 4, 8, 12, 16, 24 and 32 kHz between the control group (*n* = 18) and noise group (*n* = 18). (**b**) Myosin7a (red) and phalloidin (green) immunofluorescence staining of mice cochleae in the control group (*n* = 3) and noise group (*n* = 3). (**c**) Quantification of outer hair cells (OHCs) in the apex, middle and base segments of cochlea in the control group (*n* = 3) and noise group (*n* = 3). Scale bars = 20 μm. Data are represented as mean ± SEM. ** *p* < 0.01, *** *p* < 0.001.

**Figure 3 ijerph-19-00382-f003:**
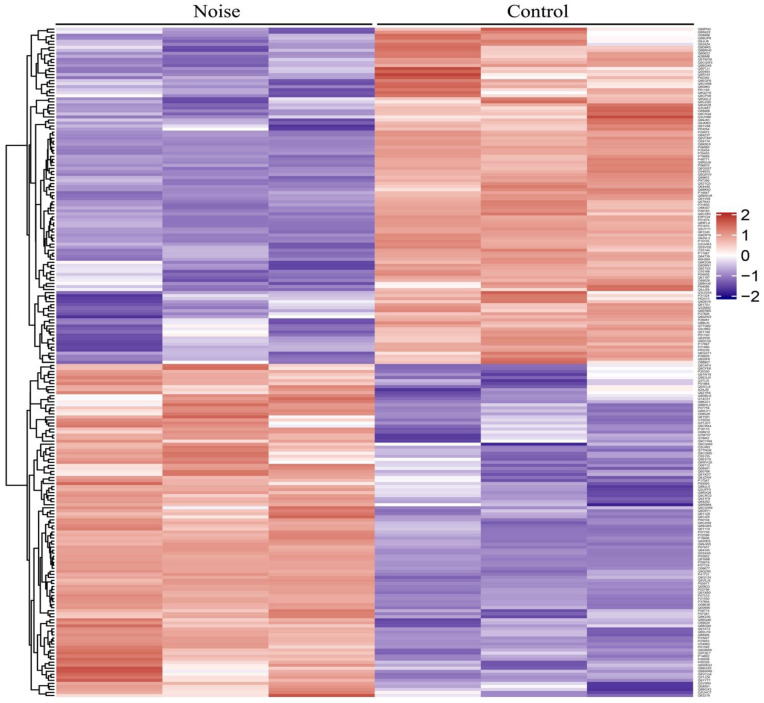
Heat map of protein abundance differences between control group and noise group. Red color indicates high abundance, and purple color indicates low abundance.

**Figure 4 ijerph-19-00382-f004:**
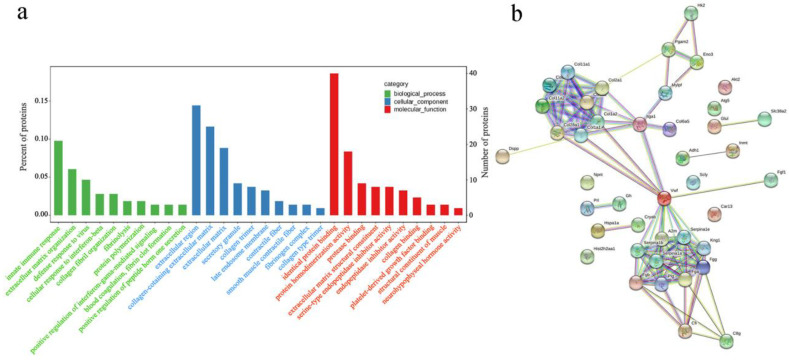
Bioinformatics analysis of identified differentially expressed proteins (DEPs). (**a**) The top 30 Gene Ontology (GO) terms of DEPs based on biological process, cellular component, and molecular function. (**b**) Protein–protein interaction (PPI) network analysis of DEPs involving in the significantly enriched signaling pathways.

**Figure 5 ijerph-19-00382-f005:**
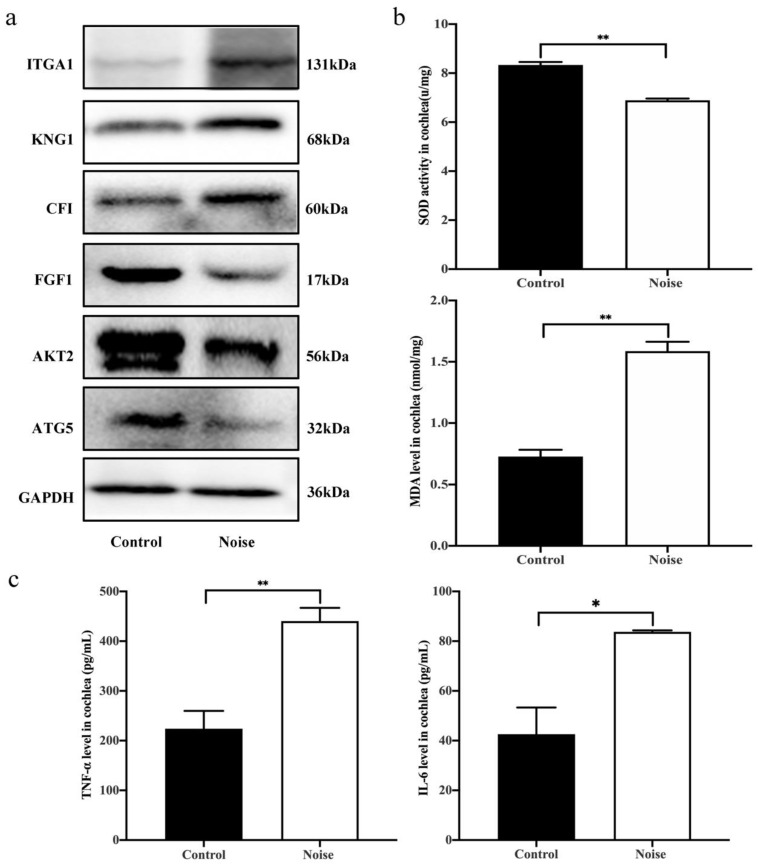
Measurement of expression levels of DEPs, oxidative stress markers and proinflammatory cytokines production. (**a**) Validation of representative inflammation and autophagy-related DEPs by western blots in the control group (*n* = 3) and noise group (*n* = 3). (**b**) Comparison of MDA level and SOD activity in cochlea between the control group (*n* = 3) and noise group (*n* = 3). (**c**) Comparison of proinflammatory cytokines TNF-α and IL-6 in cochlea between the control group (*n* = 3) and noise group (*n* = 3). Data are represented as mean ± SD. * *p* < 0.05, ** *p* < 0.01.

**Table 1 ijerph-19-00382-t001:** The significant signaling pathway involved by DEPs identified by KEGG pathway enrichment analysis.

KEGG Pathway	Protein	Count	*p* Value
PI3K/AKT signaling pathway	Col6a5, Prl, Gh1, Col1a1, Col2a1, Fgf1, Col1a2, Col9a2, Akt2, Itga1, Vwf	11	0.033
ECM-receptor interaction	Col6a5, Col1a1, Col2a1, Dspp, Col1a2, Col9a2, Npnt, Itga1, Vwf	9	9.34 × 10^−6^
Focal adhesion	Col6a5, Col1a1, Col2a1, Col1a2, Col9a2, Akt2, Mylpf, Itga1, Vwf	9	0.005
Protein digestion and absorption	Col6a5, Col1a1, Col2a1, Col1a2, Col9a2, Col28a1, Col5a2, Col11a1, Col11a2, Slc38a2	10	2.11 × 10^−6^
Complement and coagulation cascades	Fga, A2m, Fgb, Fgg, Kng1, Serpina1a, Plg, Serpina1b, Serpina1e, Cfi, Vwf, C8g	12	1.84 × 10^−8^
Platelet activation	Fga, Col1a1, Col1a2, Akt2, Fgb, Fgg, Vwf	7	0.004
Neutrophil extracellular trap formation	Fga, Akt2, Hist2h2aa1, Fgb, Fgg, Vwf	6	0.041
Nitrogen metabolism	Glul, Ca13	2	0.033
Longevity regulating pathway	Akt2, Atg5, Hspa1a, Cryab	4	0.020
Selenocompound metabolism	Scly, Inmt	2	0.033
Glycolysis/Gluconeogenesis	Hk2, Pgam2, Adh1, Eno3	4	0.022

**Table 2 ijerph-19-00382-t002:** Representative inflammation and autophagy-related DEPs involving in NIHL identified by tandem mass tag (TMT)-labeling quantitative proteomic analysis.

Protein Accession	Protein Name	Protein Description	Molecular Weight (kDa)	*p* Value	Fold Change
Q3V3R4	ITGA1	Integrin alpha 1	130.81	0.013	1.22
O08677	KNG1	Kininogen 1	73.10	2.50 × 10^−5^	1.28
Q61129	CFI	Complement factor I	67.26	0.002	1.34
P61148	FGF1	Fibroblast growth factor 1	17.42	0.017	0.83
Q60823	AKT2	RAC-beta serine/threonine-protein kinase	55.74	0.001	0.80
Q99J83	ATG5	Autophagy protein 5	32.40	0.019	0.78

## Data Availability

The datasets used and analyzed during the current study are available from the corresponding author on reasonable request.

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
