# Peer review of "TMT-Based Quantitative Proteomics Reveals Cochlear Protein Profile Alterations in Mice with Noise-Induced Hearing Loss"

_ijerph, 2021, doi:10.3390/ijerph19010382_

Round 1
Reviewer 1 Report
The manuscript, “TMT-based quantitative proteomics reveals cochlear protein profile alterations in mice with noise-induced hearing loss”, describes changes in protein expression following an acute response of a small group of mice to a single series of 120 dB noise. How many mice were used in the study? : adding up what I see from the different experiments suggests treatment of 12 mice (3 for HC histochem, 6 for proteomics, and 3 for cytokines/SOD/westerns) and 12 controls. Were all of these treatments done at the same time – was there an attempt to replicate the experiment on different days?
Why were mice exposed to 120 dB sound for 4 h when the referenced article suggested 2 h was sufficient? How much time was between the 120 dB exposure and the “hearing tests” – and how long to sample collection? This seems to be an acute response and therefore it is important to know how long after exposure changes in protein expression were measured. (It would be interesting to see different times to see if the changes in protein expression were dynamic.) The reported differences in protein levels between treated and control mouse extracts were quite small (1.2 fold) and the p values indicated for the families in Table 1 are rather large (0.03 for several) for a sample of more than 4000 different proteins. Some more details on the method of determining differences in expression and statistical analysis of the TMT-MS methodology needs to be added – the methods only mention t-tests which do not seem appropriate for these large datasets which seem to be done as a screen with an N=1.
For the SOD, MDA and westerns: were the 3 mice used for all of these? Were the samples from the 3 combined or were they assessed independently? What are the fold differences for these measures? Are the westerns an N of 1 or were multiple independent experiments assessed?
Were the differences in SOD, MDA, IL-6 and TNF shown in the independent assays also shown in the TMF-MS? What does this say about the TMT-MS? How many of the proteins indicated by TMF-MS were examine by western – just the 6? (What were the fold differences for these?) What was the false discovery rate (positive and negative) of the TMT-MS methodology?
The discussion is MUCH too long. From my perspective, it appears that the discussion is used to verify the selected candidates chosen to assess the TMT-MS (rather than using TMT-MS to pick candidates for the westerns). The TMT-MS screening method seems to be appropriate to identify candidates in a single experiment. However, I would like to see a much broader and more rigorous attempt to verify these candidates by western (or ELISA) to show that the screen identified reliable and reproducible changes in the expression of specific proteins. This means evaluation of a larger set of proteins in independently repeated experiments to confirm these changes. The fact that the TMT-MS detected changes in protein levels is so small and the statistical methods somewhat weak do not strongly support that the identified changes are reliable. It seems that literature reports, rather than experimental data, is used to support an N=1 screen.
Reviewer 2 Report
- Would you mind adding a figure or a chart showing the experimental flow? Such a figure/chart should address the aspect of time (between the steps and time spent on the experimental steps) and the number of animals/ears per experimental condition.
- As far as I understood, the ABR measurements were performed immediately after the four hours of noise exposure. The protein extraction/preparation was done as the next step after ABR. However, the surface preparations were done three days after noise exposure. Please explain.
- The conditions used during the noise exposure are not clear. How many animals were exposed at a time? Were the animals anesthetized, or were they awake? If the animals were awake, could they eat and drink? What was the minimal and maximal distance between an animal and the speaker?
- Line 122: “The numbers of immune-stained outer hair cells…” - the Authors used phalloidin and not the antibodies for staining. Thus, the cells were not "immune-stained".
- Would you please provide the catalog numbers of the reagents used? I am referring to the paragraph – the reagents in question are underlined:
- “… 0.1 M phosphate-buffered saline (PBS), and then 4% paraformaldehyde was perfused and incubated at 4 ℃ The cochleae were washed in PBS and decalcified by 10% EDTA for 3 days at 4 ℃. After incubation in a 10% goat serum for blocking nonspecific antibody binding overnight at 4°C, cochleae were incubated at 4 ℃ overnight in the darkness using an anti-Myosin7a antibody (Abcam, MA, USA) at 1:100 dilution. After washing in PBS, cochleae were incubated with phalloidin (Life technology, CA, USA)…”
- How much time has passed between finishing the noise exposure and the decapitation? Was that time window the same in all experiments? This is very important to help understand the changes in protein concentration.
- In the figure description, please indicate how many animals were used in each experiment.
- The discussion should contain self-critique relating to the experimental procedures, timing (VERY IMPORTANT) – the authors determined the concentration of proteins very shortly after noise exposure. On top of it, they did not determine what was happening in the secretome.
Reviewer 3 Report
This is an experimental study using an animal model of noise-induced hearing loss and proteomic to reveal potential biomarkers of noise induced hearing loss.
The authors use ABR, immunofluorescence in whole mount preparation, tandem mass tag labelling and liquid chromatography/ mass spectrometry to quantify proteins with bioinformatics, oxidative stress markers and proinflammatory cytokines in a male mouse model to assess the effect of noise-induced hearing loss.
The results are clearly present and the study was primarily designed for proteomic analysis. The authors reports that TNFa and IL6 were significantly increased after noise exposure. They also present a set of DEPs related to autophagy and inflammation including three upregulated DEPs (ITGA1, KNG1 and CFI) and three downregulated DEPs (FGF1, AKT2, and ATG5) were involved in PI3K/AKT pathway, ECM-receptor interaction, and focal adhesion pathway.
Tables and figures have good quality
The cytokines results should be compared with other studies in mouse and human showing increase of proinflammatory cytokines.
The study is consistent with findings previously. Novelty is medium to low.
This reviewer consider that IJER-PH is not the best journal for this type of study. Biomedicines or Biomolecules seems to be a better option. However, the quality is good enough and deserves its publication.
The authors use acronyms in the introduction such as MDA, SOD without detailing them.
Section 3.2 in results DEG should be DEP
Round 2
Reviewer 1 Report
I thank the authors for their response to my previous review. Some areas were clarified although some issues still remain.
- How much time was between noise exposure and decapitation of mice for protein preparation? The authors indicate that about 10 min were between noise exposure and ABR but the text indicates that the animals were “immediately decapitated” after ABR for proteome analysis. If this were true, then the changes in protein expression shown by the MS would have to have occurred within 15 minutes of noise exposure. This clearly is not enough time for transcription and/or translation and suggests some other mechanisms would have to be invoked. In that light, the time from noise exposure to production of SOD and MDA, and for the inflammatory cytokines also needs to be specifically described. (The time from noise exposure to hair cell IHC was described as three days - which makes sense for that particular assay.) The difference in timing interval from noise exposure to detection of protein changes is critical for understanding what is going on.
- The use of Protein Pilot to assess changes in peptide levels is completely appropriate for the screen – and the added description of some of the software option set points was useful. However, in my opinion the fold differences in peptide levels were rather small (+/- 1.2 fold) and the statistical criteria (p<0.05) were not very stringent. The KEGG pathway analysis is supportive of the candidates but such screens are notorious for producing spurious candidates. Therefore, I think a more in depth confirmation of the screening data would be useful: 6 identified proteins were examined by western and four proteins (SOD, MDA, IL-6, and TNF) which might be mechanistically linked were also examined. The changes in the 6 identified proteins roughly corresponded to the changes in the MS dataset (but this represents the tiniest fraction of the putative proteins). Did the changes in the 4 mechanistically linked proteins also correspond to the MS data? Were IL-6 peptides shown to be altered in the MS dataset – and SOD, MDA, and TNF? If not, then the analysis is susceptible to false-negative detection of candidates. This is important.
- Put the fold changes and number of independent determinations for the protein levels by western in the manuscript. (A single western for each candidate is not reliable.)
- The Discussion section is still much too long. The discussion is being used to post hoc support the peptide candidates based on previous published data rather than to discover anything new. I understand why it was done – but it should be done with many fewer words. What NOVEL candidates, or advances, were made?
- THe results are based on a single animal experiment with multiple endpoint tests. This seems to be becoming more common in publications. I always ask if the experiment was repeated at a different time because single experiments can be affected by unappreciated external stimuli which would not be seen in a single exposure. This is my bias and I see it as one way to increase reproducibility in published findings.
Reviewer 2 Report
I appreciate the detailed answers of the Authors to my previous comments. Through the introduced revisions, the manuscript gained clarity.
The English still needs corrections (e.g., "AKT2 and ATG5 were lowly expressed": should be "the expression of AKT2 and ATG5 decreased" or "was lower than in the controls". Another example "The results of western blotting" should be "Western blotting". And another example "Because high-intensity noise exposure causes oxidative stress and triggers inflammation in cochlea, thus damaging sensory HCs." - that sentence should be entirely revised.)
In addition, I explicitly asked the authors to present the experimental flow, the times used for and between the individual steps, and the numbers of animals used in each type of experiment. That has only partially been done in Figure 1. The figure lacks the times (except for the noise exposure) and the number of mice used.
Round 3
Reviewer 1 Report
The manuscript is designed to describe an MS-based approach for identifying candidate peptides that are differentially expressed between control mice and mice exposed to 120 dB sound. This screen describes over 4000 candidates - but the screen has not been adequately verified by independent means. Only 6 candidates were confirmed by western blotting. Four additional candidates that were previously identified, SOD, MDA, IL6, and TNF were apparently not identified by the MS screen - and are false negatives.
In my opinion, a number of proteins - including those upregulated and those unaffected in the MS screen - need to be independently tested to demonstrate that the MS screen actually generates a rational group of candidates. Without this characterization, there is no way to determine if the method has generated anything of value.
Author Response
Point 1: The manuscript is designed to describe an MS-based approach for identifying candidate peptides that are differentially expressed between control mice and mice exposed to 120 dB sound. This screen describes over 4000 candidates - but the screen has not been adequately verified by independent means. Only 6 candidates were confirmed by western blotting. Four additional candidates that were previously identified, SOD, MDA, IL6, and TNF were apparently not identified by the MS screen - and are false negatives.
Response 1: We sincerely appreciate your valuable comments. In our previous studies, we found that inflammation-related gene polymorphisms are significantly associated with the susceptibility to NIHL, showing that inflammation is an important stress to the pathogenesis of NIHL (PMID: 33067783; PMID: 33768461). Moreover, our previous plasma metabolomic profiling in workers with NIHL study revealed that autophagy may play a key role in the occurrence and development of NIHL (PMID: 34275074). Furthermore, recently, many studies on hearing loss have found that inflammation and autophagy are important pathogenic mechanisms (PMID: 28492547; PMID: 33187328; PMID: 23727008; PMID: 34006186). Therefore, among the proteins identified by the current proteomic analysis, we pay close attention to the inflammation- and autophagy-related proteins and selected them as biomarkers for subsequent validation. Finally, six related proteins were enrolled and verified by western blotting. Meanwhile, this is also a preliminary exploratory study, which aims to provide meaningful clues for the future studies in this field.
As for SOD, MDA, IL-6 and TNF-α, our aim was to explore the occurrence of oxidative damage and inflammation by determining its levels using the ELISA kits. Many studies showed that oxidative stress and inflammation were key factors contributing to NIHL (PMID: 25694169; PMID: 15306254; PMID: 26520584; PMID: 26776972). Hence, we detected its levels to investigate the oxidative damage and inflammation in the cochlea. Furthermore, a great number of studies have revealed that the levels of SOD, MDA, IL-6 and TNF-α in cochlea were significantly changed following noise exposure (PMID: 19051071; PMID: 31731459; PMID: 30142495; PMID: 32692754). Our present results were consistent with these findings and confirmed these significantly changes.
Proteomics is crucial for early disease diagnosis, prognosis and to monitor the disease development. It also has a vital role in drug development as target molecules (PMID: 28087761). MS is used to measure the mass to charge ratio (m/z), therefore helpful to determine the molecular weight of proteins. For MS, the proteins are extracted from the sample and digested using one or several proteases to produce definite set of peptides (PMID: 22324799). Further steps including enrichment and fractionation can be added at protein or peptide level to decrease the complexity of sample or when the analysis of specific subset of proteins is desired (PMID: 22226769; PMID: 21080488; PMID: 25033288). The obtained peptides are analyzed by liquid chromatography coupled with mass spectrometry (LC–MS).
Why were SOD, MDA, IL-6, and TNF-α not apparently identified by the MS screen, the following reasons can explain this phenomenon. There are thousands of proteins in the sample, MS could not completely detect them, this is a common phenomenon. TMT quantifies proteins according to the reporter ion peak of secondary mass spectrometry; however, reporter ion peak of some proteins is low ionization efficiency, which causing them are not easy to be detected by MS. In contrast, the ELISA kit is highly sensitive immunoassay and widely used for diagnostic purpose. The sensitivity of ELISA kit is significantly higher than MS. ELISA could be amplified by specific antibody, with a certain amplification effect, but MS directly detects ion content, without amplification effect. Therefore, this reason may explain this abnormal phenomenon and better help us understand why SOD, MDA, IL-6, and TNF-α were not detected by MS.
For this result, we have also consulted engineers and researchers engaged in proteomics, and they told us that this is a common phenomenon and they have encountered the same phenomenon in the process of their research.
Point 2: In my opinion, a number of proteins - including those upregulated and those unaffected in the MS screen - need to be independently tested to demonstrate that the MS screen actually generates a rational group of candidates. Without this characterization, there is no way to determine if the method has generated anything of value.
Response 2: We appreciate your good comments and suggestions. It has been reported that proteomics is one of the most significant methodology to comprehend the gene function (PMID: 11237011). Proteomics has powerful capacity to identify the potential biomarkers and therapeutic targets and it has been widely used to screen biomarkers for various diseases. Proteomics has been performed to explore differentially expressed proteins in various diseases, such as cancers, dengue fever, Alzheimer's disease, and pulmonary fibrosis induced by PM2.5 (PMID: 33176066; PMID: 23665002; PMID: 31485295; PMID: 31921022; PMID: 27381087; PMID: 30602677). Proteomics has been widely and successfully applied to various studies by researchers, suggesting that its important role in disease research, such as detection of various diagnostic markers, candidates for vaccine production, understanding pathogenicity mechanisms, alteration of expression patterns in response to different signals and interpretation of functional protein pathways in different diseases. Taken together, proteomics is an effective tool for early disease diagnosis, prognosis, and disease development monitoring. Therefore, in our study, the application of proteomics is no problem. As for the identified proteins, in the future, we will perform multiple independent experiments to confirm the results obtained from MS and to reveal the real changes of candidate proteins.
